# Food-Induced Brain Activity in Children with Overweight or Obesity versus Normal Weight: An Electroencephalographic Pilot Study

**DOI:** 10.3390/brainsci12121653

**Published:** 2022-12-02

**Authors:** Christine Kösling, Lisa Schäfer, Claudia Hübner, Caroline Sebert, Anja Hilbert, Ricarda Schmidt

**Affiliations:** Integrated Research and Treatment Center Adiposity Diseases, Behavioral Medicine Research Unit, Department of Psychosomatic Medicine and Psychotherapy, Leipzig University Medical Center, 04103 Leipzig, Germany

**Keywords:** brain activity, childhood obesity, EEG, food, frequency bands

## Abstract

Background: Although increased food cue reactivity is evidenced to be crucial to the development and maintenance of pediatric obesity, virtually nothing is known about the underlying neurophysiological aspects of food cue reactivity in children with obesity. Therefore, this study aimed at investigating neural characteristics in children with overweight or obesity using electroencephalography (EEG). Methods: Electrophysiological brain activity was measured using EEG frequency band analysis in *n* = 9 children with overweight or obesity versus *n* = 16 children with normal weight (8–13 years) during the presentation of high- and low-calorie food pictures and images of appealing non-food stimuli. Results: Children with overweight or obesity showed significantly increased relative central beta band activity induced by high-calorie foods and appealing non-food stimuli compared to children with normal weight. Beyond significant effects of the scalp region on EEG activity, non-significant effects of stimulus category or weight status were seen for theta and alpha frequency bands. Conclusions: This study demonstrated elevated beta band activity in children with overweight or obesity when viewing high-calorie food stimuli. Beta band activity may, thus, be a valuable target for neuromodulatory interventions in children with overweight or obesity.

## 1. Introduction

In the last decades, the global prevalence of obesity in children and adolescents (body mass index [BMI, kg/m^2^] percentile > 97) has risen considerably [1], having now stabilized at a fairly high level in developed countries [2,3]. This is particularly serious considering its high chronicity [4], premature mortality [5], and its physical comorbidities, which include hypertension, metabolic disorders [6], and mental disorders such as attention-deficit/hyperactivity disorder (ADHD), depression, and conduct disorders [7]. Notably, the recommended multimodal weight-loss programs, consisting of dietary interventions, physical activity, and behavioral therapy [8], were found to have small effects on weight loss at best [9].

Emerging evidence has shown that obesity may be linked to changes in physiological and psychological processing of food cues, specifically high food-cue reactivity, which describes physiologically conditioned responses to food usually expressed by increased neural activity, salivation, or physiological arousal. Psychologically, these responses are reliably experienced as a conscious and intense desire to eat, or as food cravings [10]. Meta-analytic data indicated medium-to-large effects for food-cue reactivity to cross-sectionally and longitudinally predict overeating, weight gain, and overweight and obesity in children and adults [10]. Much research operationalized cue reactivity in response to food stimuli in children via self-reported hunger and craving levels [11,12] as well as food intake [13,14,15]. However, neural measures are evidenced to be more direct and stronger indicators of food-cue reactivity than behavioral measures [10]. 

So far, neural food-cue reactivity in children and adolescents was most commonly derived from functional magnetic resonance imaging (fMRI). Several neuroimaging studies using fMRI revealed automatic recruitment of brain regions related to the reward and motivation system (e.g., insula, orbitofrontal cortex [OFC]), metabolic signaling (e.g., hypothalamus), emotion (e.g., amygdala), and attention and visual processing (e.g., occipital cortex and parietal cortex) when children were shown food stimuli in a fasting state, without effects on children’s BMI [16,17,18,19]. These effects thus mirror normal motivational responses to food cues in states of hunger, and tend to decrease with higher levels of satiety [18,20]. Strikingly, in children with obesity, neural food-cue reactivity was shown to fail to diminish after food intake compared to children with normal weight, particularly in the OFC [16], which may depict a pathological hyper-responsiveness to food cues. Indeed, studies in children that experimentally established satiety before scanning showed that children with obesity had greater OFC activity than children with normal weight when passively viewing high-calorie food images [21,22,23], while for regions related to cognitive control (e.g., dorsolateral prefrontal cortex [DLPFC]), mixed evidence has been found [22,23,24].

Beyond fMRI, which is based on hemodynamic responses (i.e., changes in blood flow), food-specific neural mechanisms may be derived from electroencephalography (EEG), a technique that allows the visualization of electrophysiological processes in the brain. It is a non-invasive method with high temporal resolution, in which the electrical activity of the neurons is derived in the form of potential differences on the scalp, and, thus, is a more direct measure of cortical activity than fMRI. Depending on the frequency of the voltage fluctuations, different frequency bands are distinguished, ranging from delta (1.0–3.5 Hz) to beta (12.5–30.0 Hz), and depicting different cognitive states. For example, theta waves indicate states of cognitive control and alpha waves mirror relaxed waking states, while beta activity is considered as a marker of focused thinking and directed attention. Various mental disorders are characterized by alterations of EEG-measured brain activity, especially those with deficits in self-regulation, as seen in ADHD [25]. The few studies using quantitative EEG in individuals with overweight or obesity proposed a neurophysiological basis of obesity in children and adults during a resting state [26,27,28,29], during neuropsychological tasks [30,31,32,33], and while perceiving disorder-specific stimuli [29,34,35]. In resting state, children as well as adults with overweight or obesity showed reduced alpha band activity, likely reflecting abnormal arousal and vigilance processes [26,28]. While deviant alpha band activity in adults with overweight or obesity has been shown for parietal, temporal, and occipital areas [26], no regional differences were described in children [28]. Furthermore, adults with obesity and comorbid binge-eating disorder presented increased beta band activity in resting state conditions, especially in the fronto-central brain regions [29,34]. Deviations in frontal beta band activity were also observed in adolescents and adults with overweight or obesity during cognitive tasks [30], involving both neutral and high-calorie food pictures [31,32], which may reflect increased attention and vigilance towards these stimuli. 

Only a few EEG studies addressed electrophysiological parameters of food-cue reactivity in individuals with obesity based on frequency band analysis. Following a single taste of a chocolate milkshake, adult patients with overweight or obesity and symptoms of food addiction showed increased delta and theta band activity in the right hemisphere during states of hunger, primarily in the frontal and precentral gyrus and in the insula, compared to controls [35]. Group differences in EEG activity were not seen in a neutral solution condition. Tammela et al. measured fronto-central brain activity in women with obesity with and without binge-eating disorder during states of hunger [29]. Participants were presented a warm, freshly cooked meal in the experimental condition and an image of a landscape, which served as the control condition. In both groups, the fronto-central beta band activity was higher in the food than in the control condition, pointing to greater alertness and vigilance to food compared with control stimuli in women with obesity [29]. The apparent difference in EEG activity between these two studies, i.e., deviant slow-wave [35] versus fast-wave activity [29], may be related to different types of food presentation (ingesting real food versus visual stimuli) or different comparators, which warrants further research. 

### Aims and Hypotheses

In summary, the available evidence supports the assumption that there are obesity-specific deviations in brain activity, although evidence is still little in this respect, and the data are not yet comprehensive, especially in children. In addition, due to different formats and tasks, the studies are difficult to compare. Specifically, there is a lack of a systematic investigation of EEG activity with frequency band analysis in response to visual food stimuli compared to adequate control stimuli in states of satiety, where deviant EEG patterns would more precisely indicate pathophysiological processes than during states of hunger. In order to evaluate children’s neural reactivity to food stimuli, this study assessed electrophysiological processing of high- and low-calorie food stimuli in comparison to appealing non-food stimuli in children in a satiated state. Based on extant evidence, we hypothesized an increase in frontal and central beta band activity and a reduction in parietal, temporal, and occipital alpha band activity only during the high-calorie food condition in children with overweight or obesity versus children with normal weight. With regard to the theta frequency band, data analysis was exploratory, as no directional hypothesis was formulated due to the rare and partly inconsistent findings. Regarding the type of stimuli, we expected that group differences would occur for high-calorie foods, but not for low-calorie foods and neutral stimuli.

## 2. Materials and Methods

### 2.1. Participants and Study Design

Children of the age of 8 to 13 years were recruited from the population through advertisements in digital media, supermarkets, and citizen’s offices. In addition, study flyers and posters were placed in clinical institutions (e.g., University Medical Center, local pediatricians, outpatient weight-reduction programs). Interested children and their parents contacted the study team via telephone or email, and an appointment for a telephone screening was scheduled to evaluate inclusion and exclusion criteria. The prerequisite for inclusion in the group with overweight or obesity (OO) was objectively measured overweight or obesity using calibrated scales, defined as a BMI percentile (BMIP) > 90 based on age- and sex-specific German reference data [36]. The control group of children with normal weight (NW), which was stratified to the OO group based on age and sex, required a BMIP within the normal weight range (10 ≤ BMIP ≤ 90). Exclusion criteria for both groups were left-handedness; a neurological or mental disorder, including ADHD and eating disorders; or a severe physical disease. Further exclusion criteria were medication intake with effect on brain activity, executive functions, or body weight, and mental retardation of the child. Each participant received financial compensation of EUR 15 in total. 

The OO group consisted of *n* = 12 children (*n* = 5 children with overweight [90 < BMIP ≤ 97] and *n* = 7 children with obesity [BMIP > 97]), while *n* = 22 children with normal weight were included in the NW group. Due to data loss caused by a technical failure, *n* = 2 children with overweight, *n* = 1 child with obesity of the OO group and *n* = 6 children of the NW group had to be excluded, leaving a total sample of *N* = 25 with group sizes of *n* = 9 and *n* = 16 for the OO and NW groups, respectively. The total sample was 10.83 (±1.79 SD) years old, included slightly more boys (*n* = 14, 56%) than girls (*n* = 11, 44%), as well as more families with high socio-economic status (*n* = 15, 60%), as measured by the Winkler index [37]. All children were of German nationality. The children’s mean BMIP was 70.42 (±25.92 SD), with values of 96.8 (±3.2 SD, range: 91–100) for the OO group and 55.6 (±20.5 SD, range: 16–82) for the NW group. Apart from the different weight status, there were no other significant differences regarding socio-demographic parameters, including socioeconomic status (see Table 1).

### 2.2. Procedure

Children and their parents were contacted on the phone to screen for inclusion and exclusion criteria (*n* = 62). Eligible children (*n* = 34) were then invited to the laboratory. Children were instructed to eat 1 h before coming to the laboratory in order to establish satiety among participants. Although there was no time window for testing specified, most assessments took place in the afternoon (*n* = 21, 84%). The OO and NW groups did not differ in the median time of testing (2.30 pm in both groups, *p* > 0.05). Time of testing was not associated with levels of hunger in the total sample (*r* = 0.056, *p* = 0.792). The experimental sessions were conducted by two trained Ph.D. and M.D. students according to a standardized procedure. The practical portion began with a matrices test to estimate children’s non-verbal fluid reasoning (see below), followed by EEG recordings. During electrode montage, children completed self-report questionnaires, including a rating of their current hunger level and the desire to eat. After the EEG recording, children completed the same ratings once again before taking a 5 min break. Clinical interviews were then conducted with the child and one parent to evaluate the presence of eating (Eating Disorder Examination adapted for Children) [38,39] and mental disorders (Diagnostic Interview for Mental Disorders in Children and Adolescents) [40]. Finally, anthropometric data were measured. Altogether, the experimental session lasted between 2.5 and 3 h.

### 2.3. EEG Recordings

EEG recordings took place in a darkened, acoustically and electromagnetically shielded cabin with video control. A 32 channel QuickAmp amplifier and BrainVision Analyzer 2.1 software were used (Brain Products, Gilching, Germany). EEG signals were recorded over 19 scalp sites (Fp1, Fp2, F3, F4, F7, F8, Fz, C3, C4, Cz, T3, T4, T5, T6, P3, P4, Pz, O1, and O2) at a sampling rate of 250 Hz. Ag/AgCl electrodes were positioned according to the standard 10–20 International system and referred to the right mastoid. In order to exclude artifacts caused by eye movements and pulse beats, the electrooculogram and the electrocardiogram were also derived. Impedances were kept below 5 kΩ before starting the recording. Children sat 1.5 m in front of a screen on which the experimental pictures were shown, and were asked to look at the pictures attentively. 

### 2.4. EEG Stimuli

The stimuli were 20 images of high-calorie foods (e.g., cake, hamburger, chocolate), 20 images of low-calorie foods (e.g., strawberry, cucumber, watermelon), and 20 images of appealing neutral stimuli (e.g., kitten, bicycle, emoji) presented in three separate blocks. Each image was shown for 5 s; thus, each block lasted 100 s. The order of the blocks and of the pictures within the blocks was randomized, with a fixation cross between the blocks for 10 s, see Figure 1. The food pictures were chosen from a large image database for experimental research on eating and appetite [41]. This database contains food images and non-food images with detailed meta-data, including ratings of recognizability, palatability, macronutrients, and image characteristics. Pictures of neutral stimuli were also selected from this database and supplemented with images from an online search. The final image selection was based on a pre-study, in which *n* = 38 primary school children aged between 9 and 12 years evaluated 80 pre-selected pictures with high ratings of recognizability and familiarity. According to children’s ratings of ‘recognizability’ and ‘liking’, the 20 pictures with the highest ratings in each category (high-calorie, low-calorie, and pleasant non-food items) were selected for the main study. For better comparability, we selected images of single foods only, but no images of prepared meals. The classification into high- and low-calorie was made on the basis of the energy density of the food: foods with an energy density of ≥2.5 kcal/g were considered high-calorie, foods with ≤1.5 kcal/g were classified as low-calorie.

### 2.5. Pre-Processing and Analysis of EEG Recordings

A band-pass filter of 0.53–70 Hz and a notch filter of 50 Hz were used to filter EEG data. The three EEG blocks of 100 s each were segmented into 1 s intervals. The ocular artifact correction was conducted in accordance with the Gratton et al. algorithm [42], and an automatic artifact rejection was applied to segments with voltage steps >50 µV/ms and amplitudes exceeding ±100 µV. In addition, the segments were manually analyzed by two trained researchers to filter out remaining impurities such as acoustic disturbances, movement artifacts, and focal abnormalities. At least 30 s of EEG artifact-free recording (not necessarily consecutive) was analyzed for each participant and each condition. The average number of remaining segments was 65.5 (±22.0 *SD*) for high-calorie, 64.9 (±21.9 *SD*) for low-calorie, and 60.8 (±20.4 *SD*) for neutral images. 

The filtered EEG data were Fourier transformed with a Hanning window length of 20% extracting theta (θ, 3.5–7.5 Hz), alpha (α, 7.5–12.5 Hz), and beta (β, 12.5–30 Hz) frequency bands. The extracted absolute power for each frequency band was grouped for frontal (Fp1, Fp2, F3, F4, F7, F8, Fz), central (C3, C4, Cz), temporal (T3, T4, T5, T6), parietal (P3, P4, Pz), and occipital (O1, O2) regions, converted to relative band power (%), and ln-transformed to obtain normally distributed data.

### 2.6. Control Variables

Both shortly before and immediately after the EEG recording, participants’ hunger and mood were evaluated. Specifically, the level of hunger, tiredness, and boredom, as well as the desire to eat, were assessed by means of a self-report questionnaire, with response options ranging from 1 = not at all to 7 = very strong. Furthermore, an evaluation of the images was performed after the EEG recording, in which the children were asked to rate all previously shown pictures for recognizability, liking, and desire to eat (except for neutral stimuli) on a scale ranging from 1 = very good/very much to 6 = very bad/not at all. Children’s level of non-verbal intelligence was estimated using the ‘matrix reasoning’ subtest of the Wechsler Intelligence Scale for Children—Fourth Edition (WISC-IV) [43], with age-standardized values ranging from 1–19 and higher scores indicating higher intelligence. 

### 2.7. Data Analysis

First, data were evaluated for group differences in socio-demographic variables, estimates of non-verbal intelligence, as well as pre-EEG ratings on hunger and mood, to identify significantly different variables between groups or those with significant effects to be included as covariates in the main analysis. Second, to evaluate effects of scalp region, group, and condition on EEG frequency band activities, generalized linear models with maximum likelihood estimation were run with the factors region (frontal, central, temporal, parietal, occipital), condition (high-calorie, low-calorie, neural), group (OO, NW), and Region × Condition × Group interaction. The pre-EEG rating of desire to eat served as a covariate in the model. Pairwise post hoc comparisons were made and reported in case of significant higher-order effects. 

Effect sizes were interpreted according to Cohen (1988), using Cramer’s *V* and Hedges *g,* with values *V* = 0.10, *g* = 0.20 referring to small, *V* = 0.30, *g* = 0.50 to medium, and *V* = 0.50, *g* = 0.80 to large effects. IBM SPSS Statistics (Version 24) was used for statistical analyses, and a two-tailed significance level was set at α < 0.05.

## 3. Results

### 3.1. Preliminary Analyses

For socio-demographic variables and children’s levels of non-verbal intelligence, no significant group differences were seen (see Table 1). Although both groups reported low levels of hunger and desire to eat prior to EEG recordings, the NW group showed a non-significantly greater desire to eat, with a larger effect than the OO group, but without significant effects for ratings of hunger. Regarding post-EEG ratings, both groups recognized all pictures very well, without group differences. The NW group reported a significantly higher desire to eat for low-calorie food compared to the OO group, with a large effect size. No other significant effects were seen.

### 3.2. EEG Activity 

#### 3.2.1. Theta

There was a significant main effect of scalp region on theta activity (*p* < 0.001), but no effect of condition or group, nor a three-way interaction (see Table 2 and Table 3). Post hoc tests showed significantly different theta activity between all five scalp regions, with large effects (0.93 < *g* < 9.51), with the highest values for frontal brain regions and the lowest values for occipital regions. 

#### 3.2.2. Alpha 

For alpha band activity, region had a significant effect (*p* < 0.001), indicating significantly different brain activity between scalp regions (0.42 < *g* < 4.62), except for the temporal and parietal regions (*g* = 0.09). Across groups and conditions, the highest alpha band activity was found for frontal regions, while the lowest alpha band activity emerged at occipital regions. No significant main effect for condition, group, or a three-way interaction was seen.

#### 3.2.3. Beta

There were significant main effects of region (*p* < 0.001) and group (*p* = 0.031), along with a marginally significant Region × Condition × Group interaction (*p* = 0.095) and a non-significant effect of condition. Post hoc tests showed a significantly greater beta band activity in central regions for the OO compared to the NW group for high-calorie food, *t* (23) = 2.281, *p* = 0.032, *g* = 0.95, and neutral stimuli, *t* (23) = 2.418, *p* = 0.024, *g* = 1.00, but not for low-calorie food stimuli, *t* (23) = 2.017, *p* = 0.056, *g* = 0.89. No other significant effects were seen. 

## 4. Discussion

Unlike available research in adults with overweight and obesity [34], to the best of our knowledge, this study is the first to systematically investigate food-specific neural processes in children with overweight or obesity using quantitative EEG. For this purpose, 8- to 13-year-old participants with overweight or obesity and an age- and sex-matched normal-weight control group were shown images of high- and low-calorie food and appealing neutral stimuli during EEG recording. The results indicated a significantly increased level of central beta band activity in children with overweight or obesity versus children with normal weight, both for high-calorie food and appealing non-food stimuli. Regarding alpha and theta band activity, no significant differences were found, neither in relation to the stimuli nor to BMI.

These results confirm our hypothesis of increased central beta band activity in children with overweight or obesity relative to those with normal weight, induced by high-calorie food stimuli. This finding is in line with previous studies, which have shown deviations in beta band activity in adolescents and adults with overweight or obesity during the performance of sustained attention tasks [30,31,32] and in adults with obesity while viewing food stimuli [29]. In the literature, elevated beta band activity is considered to mirror increased perceptive efforts and focus on the presented stimuli [31,44]. The deviating beta band activity shown in this study can, therefore, be interpreted in terms of increased attention to appealing stimuli in children with overweight or obesity, both in regard to food and neutral stimuli. Notably, previous studies in adults with overweight or obesity, which demonstrated altered beta band activity in response to food stimuli relative to controls, only used neutral non-food images for comparison, such as landscapes or office articles [29,31,32]. With the appropriate and stimulant control conditions, this study provided an indication that children with overweight or obesity not only show altered responses in terms of increased activation to high-calorie foods, but to rewards in general. This study thus provides evidence for a generally increased reactivity towards rewards in children with overweight or obesity. Contrary to our expectations, this study did not find differences in frontal beta band activity, which might be due to the immature frontal lobe in children compared to adults [45,46]. Keeping in mind that children’s EEG is characterized by decreased frontal and increased posterior activation compared to adults [47], findings from adult samples are hardly transferrable to children; rather, further comparative studies with participants of the same age are necessary to validate the present results.

Our hypothesis of reduced alpha band activity in children with overweight or obesity during the high-caloric food cue condition was not confirmed. This could be due to the fact that alpha band deviations described so far in children [28] and adults [26,27] with overweight or obesity have only been observed in eyes-closed, resting-state conditions. In the relaxed state with eyes closed, EEG activity is characterized by a posterior alpha rhythm that disappears as soon as the patient opens his eyes [48]. In the awake and concentrated state, alpha activity fades into the background, which may explain the lack of effects during stimulus presentation. The fact that alpha band activity was mainly pronounced in frontal brain regions and decreased towards the occipital regions is rather atypical, but again, comparative studies with stimulus presentation in children are lacking.

Regarding theta band activity, no effects of condition or group were observed either, which is consistent with the majority of evidence in adults with obesity that did not identify deviant theta band activity, either during resting state or cognitive tasks [34]. Hume et al. revealed lower theta band activity in adults with versus without obesity during a Stroop task using office stimuli, but non-significant effects for theta activity during a food-specific Stroop task [32]. In contrast, Imperatori et al. revealed increased frontal theta band activity after a single taste of a milkshake in adults with overweight or obesity and symptoms of food addiction versus weight-matched controls without food addiction [35]. Although theta band activity may be indicative of cognitive control and, thus, potentially relevant for food-cue reactivity in obesity [34], more evidence based on comparable methods is necessary to draw more specific conclusions on the role of theta band activity in overweight and obesity. The fact that low-frequency oscillations were supposed to originate in the prefrontal cortex and extend from there to various cortical and subcortical regions may explain why theta band activity was strongest frontally and decreased towards the occipital region [31,32].

A strength of this study is the accurate performance of the EEG recordings and their analysis. The recordings were conducted in an acoustically and electromagnetically shielded cabin with video surveillance, so that external interference factors on the EEG were reduced. Frequency bands were tested for artifacts, both automatically and manually, by two independent experts. In order to exclude possible interference between comorbid mental or eating disorders and brain activity, clinical interviews were performed. Finally, a preliminary study was conducted with primary school children to ensure that the items shown during the EEG recording are recognized by children of this age group. The most important limitation of this study was certainly the small sample size, which likely prevented some large- and medium-sized effects from becoming significant. In addition, some data were lost due to a technical defect, which further reduced the sample size. Therefore, the corresponding effect sizes were specified. Due to the small sample size, a differentiation of EEG characteristics between overweight and obesity was not possible. Finally, the cross-sectional nature of the study did not enable conclusions to be drawn regarding effect causality. Therefore, more longitudinal research is needed to determine the direction of the relationship between obesity and EEG alterations, and to reveal crucial neuropsychological intervention periods in the development of children.

## 5. Conclusions

In summary, children with overweight or obesity responded to both high-calorie and appealing neutral stimuli with increased beta band activity, while no significant differences were seen for alpha and theta frequency bands. This study thus indicated that children with overweight or obesity show an enhanced sensitivity not only to high-caloric food stimuli, but to attractive rewards in general. Increased vulnerability to unhealthy food stimuli in children with pathologically increased body weight is all the more serious in an environment characterized by the oversupply and constant availability of food, where a high degree of self-control is necessary to resist the impulse of overeating. Future studies may help in evaluating whether exposure to rewards in general has an effect of increasing food-cue reactivity in children with overweight or obesity. There is a need for studies with larger sample sizes in order to provide more reliable evidence on an obesity-specific EEG profile of children in a food-specific context. Common comorbidities, such as ADHD, should be systematically considered in order to examine whether they lead to an accumulation of observed effects in children with overweight or obesity. Regarding the clinical management of overweight and obesity, the further development and evaluation of neuromodulatory treatment, with a focus on food-cue reactivity, may be valuable as an adjunct to behavioral weight-loss treatment, given the neurophysiological and neuropsychological deviations. For example, EEG neurofeedback could be a promising alternative to other recently developed interventive approaches, such as attentional bias modification [49] or approach avoidance training [50], in children with overweight or obesity, in order to reduce food cue reactivity and improve food-specific self-control.

## Figures and Tables

**Figure 1 brainsci-12-01653-f001:**
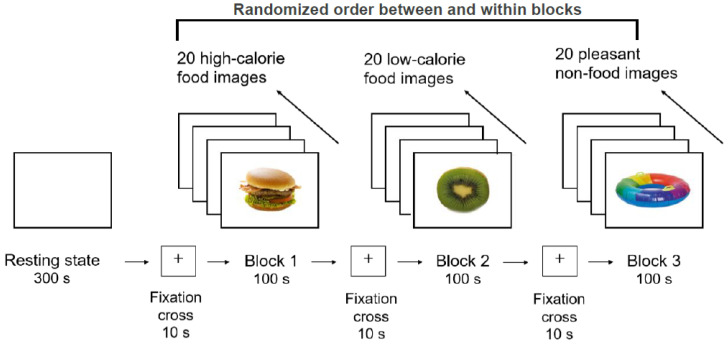
Design of the experimental procedure. After resting-state electroencephalography, three blocks of high-calorie, low-calorie, and pleasant non-food stimuli were presented in a randomized order on the screen. Images were randomly presented within a block and shown for 5 s each, adding up to 100 s per stimulus block.

**Table 1 brainsci-12-01653-t001:** Sociodemographic sample characteristics and control variables.

	OO (*n* = 9)	NW (*n* = 16)	Test Statistics		
	*M* (*SD*)	*M* (*SD*)	*t* (23)	*p*	*g* or *V*
**Sociodemographics**					
Age, years	11.18 (2.16)	10.63 (1.58)	−0.73	0.475	0.31
Sex, female, *n* (%)	5 (56)	6 (38)	Fisher’s Exact Test	0.434	0.18
SES, *n* (%)			Fisher’s Exact Test	0.397	0.24
Medium SES	5 (56)	5 (31)			
High SES	4 (44)	11 (69)			
**Anthropometrics**					
Height, cm	152.40 (14.28)	148.76 (13.44)	−0.64	0.531	0.27
Weight, kg	63.36 (21.65)	40.01 (9.42)	−3.08	0.012	1.57
BMI percentile (0−100)	96.78 (3.23)	55.60 (20.45)	−7.88	<0.001	2.48
BMI-SDS	2.15 (0.51)	0.16 (0.59)	−8.51	<0.001	3.53
**Intelligence estimate**					
Matrix reasoning (1–19)	11.22 (3.23)	12.44 (1.75)	1.05	0.319	−0.51
**Pre-EEG**					
I wish to eat (1–7)	1.33 (1.00)	2.50 (1.59)	1.98	0.060	0.83
I am hungry (1–7)	1.44 (0.73)	2.13 (1.63)	1.44	0.165	0.50
I am tired (1–7)	2.89 (1.54)	2.63 (1.86)	−0.36	0.721	0.15
I am bored (1–7)	2.56 (2.13)	2.19 (1.87)	−0.45	0.657	0.19
**Post-EEG picture rating, high-calorie**					
Recognizability (1–6)	1.26 (0.31)	1.30 (0.30)	0.28	0.780	0.13
Liking (1–6)	2.67 (1.12)	2.00 (0.97)	−1.57	0.131	0.65
Desire to eat (1–6)	3.22 (1.39)	2.44 (1.26)	−1.44	0.164	0.60
**Post-EEG picture rating, low-calorie**					
Recognizability (1–6)	1.23 (0.28)	1.27 (0.35)	0.28	0.780	0.12
Liking (1–6)	2.67 (0.87)	2.00 (0.97)	−1.72	0.100	0.72
Desire to eat (1–6)	3.67 (1.23)	2.56 (0.73)	−2.85	0.009	1.19
**Post-EEG picture rating, neutral**					
Recognizability (1–6)	1.20 (0.26)	1.27 (0.28)	0.63	0.538	0.26
Liking (1–6)	2.44 (0.73)	1.88 (0.89)	−1.64	0.115	0.67

Note. BMI, Body Mass Index; BMI-SDS, Body Mass Index-Standard Deviation Score; EEG, electroencephalography; NW, group with normal weight; OO, group with overweight or obesity; SES, socioeconomic status. Pre-EEG ratings were answered on a 7-point scale ranging from 1 = not at all to 7 = very much. Picture ratings were answered on a 6-point scale ranging from 1 = very good/very much to 6 = very bad/not at all.

**Table 2 brainsci-12-01653-t002:** Descriptive data on frequency band analyses of stimulus-elicited electroencephalography activity as a function of weight status.

		OO (*n* = 9)			NW (*n* = 16)	
	High-Calorie	Low-Calorie	Neutral	High-Calorie	Low-Calorie	Neutral
	*M* (*SD*)	*M* (*SD*)	*M* (*SD*)	*M* (*SD*)	*M* (*SD*)	*M* (*SD*)
**Theta**						
Frontal	3.54 (0.06)	3.55 (0.07)	3.55 (0.09)	3.49 (0.16)	3.50 (0.14)	3.49 (0.15)
Central	2.88 (0.09)	2.87 (0.09)	2.88 (0.11)	2.94 (0.17)	2.94 (0.16)	2.92 (0.20)
Temporal	3.09 (0.08)	3.07 (0.06)	3.03 (0.07)	3.03 (0.10)	3.02 (0.08)	3.02 (0.09)
Parietal	2.72 (0.16)	2.74 (0.16)	2.79 (0.15)	2.82 (0.10)	2.80 (0.12)	2.82 (0.10)
Occipital	2.32 (0.10)	2.30 (0.13)	2.32 (0.08)	2.31 (0.14)	2.33 (0.12)	2.34 (0.14)
**Alpha**						
Frontal	3.41 (0.10)	3.40 (0.09)	3.39 (0.08)	3.34 (0.15)	3.32 (0.16)	3.35 (0.16)
Central	2.88 (0.13)	2.89 (0.14)	2.91 (0.10)	2.97 (0.16)	2.94 (0.17)	2.97 (0.19)
Temporal	3.04 (0.10)	3.04 (0.08)	3.05 (0.08)	2.97 (0.12)	2.98 (0.14)	2.97 (0.14)
Parietal	3.00 (0.09)	2.98 (0.08)	2.98 (0.07)	3.02 (0.12)	3.03 (0.14)	3.01 (0.11)
Occipital	2.33 (0.23)	2.38 (0.24)	2.34 (0.22)	2.41 (0.30)	2.46 (0.28)	2.41 (0.28)
**Beta**						
Frontal	3.75 (0.15)	3.77 (0.15)	3.75 (0.17)	3.79 (0.17)	3.79 (0.18)	3.78 (0.19)
Central	2.61 (0.12)	2.61 (0.10)	2.60 (0.11)	2.45 (0.19)	2.45 (0.21)	2.44 (0.18)
Temporal	3.07 (0.11)	3.08 (0.13)	3.11 (0.17)	3.16 (0.17)	3.16 (0.17)	3.17 (0.20)
Parietal	2.50 (0.15)	2.47 (0.15)	2.45 (0.12)	2.39 (0.23)	2.38 (0.24)	2.37 (0.23)
Occipital	2.17 (0.33)	2.14 (0.29)	2.14 (0.31)	2.08 (0.33)	2.07 (0.34)	2.09 (0.36)

Note. Ln-transformed relative band power (%) is displayed. OO, group with overweight and obesity; NW, group with normal weight.

**Table 3 brainsci-12-01653-t003:** Electroencephalographic activity for the theta, alpha, and beta band based on scalp region, condition, group, and their interaction.

	Omnibus Test	Region	Condition	Group	Region × Condition × Group	Post Hoc Tests
	Wald χ^2^ (*df* = 29)	χ^2^ (*df* = 4)	χ^2^ (*df* = 2)	χ^2^ (*df* = 1)	χ^2^ (*df* = 22)	
Theta	906.80 ***	4525.426 ***	0.032	0.421	23.545	Frontal > temporal > central > parietal > occipital
Alpha	580.00 ***	1170.744 ***	0.068	0.168	22.578	Frontal > temporal = parietal > central > occipital
Beta	829.33 ***	3723.751 ***	0.090	4.629 *	31.033 ^†^	Frontal > temporal > central > parietal > occipital; OO > NW

Note. OO, group with overweight and obesity; NW, group with normal weight. Region: frontal, temporal, central, parietal, occipital. Condition: high calorie, low calorie, neutral. Group: overweight and obesity, normal weight. ^†^
*p* < 0.10; * *p* < 0.05; *** *p* < 0.001

## Data Availability

The data that support the findings of this study are available from the corresponding author upon request.

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
