# Peer review of "Food-Induced Brain Activity in Children with Overweight or Obesity versus Normal Weight: An Electroencephalographic Pilot Study"

_brainsci, 2022, doi:10.3390/brainsci12121653_

Round 1

Reviewer 1 Report

In this manuscript, authors compared brain activities in response to food and non-food stimuli using electroencephalography (EEG) in overweight/obese and normal weight children. “Introduction” ,“Materials and Methods” and “Discussion” provide enough background information, detailed methods and discussion.  There are several suggestions may help improving the presentation of results.

Line 95-97, description of the study is difficult for readers to read. please consider to rewrite the sentence.

Table 2. Please provide n number for each OO and NW group.

Table 3. Please define * and † here. Please consider to define or list variables for “Region”, “Condition” and “Group” in footnote.

Author Response

  1. In this manuscript, authors compared brain activities in response to food and non-food stimuli using electroencephalography (EEG) in overweight/obese and normal weight children. “Introduction” ,“Materials and Methods” and “Discussion” provide enough background information, detailed methods and discussion. There are several suggestions may help improving the presentation of results.

Response: Thank you very much. We carefully edited the manuscript to implement your suggestions and improve the manuscript, especially the presentation of results.

  1. Line 95-97, description of the study is difficult for readers to read. please consider to rewrite the sentence.

Response: Thank you for the helpful comment. As recommended, we rewrote the sentence, now stating: “Tammela et al. measured fronto-central brain activity in women with obesity with and without binge-eating disorder during states of hunger [29]. Participants were presented a freshly, warm cooked meal in the experimental condition and an image of a landscape, which served as control condition.”, lines 9599.

  1. Table 2. Please provide n number for each OO and NW group.

Response: As suggested, we added n in Table 2 to depict the sample size for each group.

  1. Table 3. Please define * and † here. Please consider to define or list variables for “Region”, “Condition” and “Group” in footnote.

Response: Thank you. As recommended, we now defined all superscripts and added information on the factors in the footnote of Table 3.

Reviewer 2 Report

This is a very interesting study investigating the neural correlates of overweight and obesity in children by means of electroencephalography. The paper is well-written and of interest for the journal. However, several minor changes should be made before considering it for publication.

Abstract.

1- Neuromodulatory interventions are recommended as beta band activity may be a potential target of treatment. This is a good conclusion. I recommend to remove the previous conclusions, for instance, those reporting that "the results indicated...".

Introduction.

1- At the end of the introduction section, the authors should describe in detail the main aims of the present study. The rationale has been well described and the hypotheses are good as well: "We hypothesized an increased frontal and central beta band activity...". However, I recommend to expand the aims. Even, I suggest to describe them into a separate subsection "1.1.".

Material and methods

1- Section 2.1. is based on the description of patients with normal weight, overweight and obsesity. Perhaps, this section should be better renamed as "Participants and study design. " More details about how were participants recruited would be useful.

2-Data Analytic Plan is the main Data Analysis. Please, rename this section.

Results

3.1. Main characteristics of the total sample should be described.

Discussion.

1-At the beginning of the discussion section, the authors report that to the best of our knowledge, this is the first study investigating neural correlated processes in children with overweight and obesity. I recommend to expand it and describe into a Strengths section.

Author Response

  1. This is a very interesting study investigating the neural correlates of overweight and obesity in children by means of electroencephalography. The paper is well-written and of interest for the journal. However, several minor changes should be made before considering it for publication.

Response: Thank you very much. We carefully edited the manuscript according to your suggestions.

  1. Abstract: Neuromodulatory interventions are recommended as beta band activity may be a potential target of treatment. This is a good conclusion. I recommend to remove the previous conclusions, for instance, those reporting that "the results indicated...".

Response: As recommended, we now omitted the sentence on increased reactivity to rewards in general from the abstract.

  1. Introduction: At the end of the introduction section, the authors should describe in detail the main aims of the present study. The rationale has been well described and the hypotheses are good as well: "We hypothesized an increased frontal and central beta band activity...". However, I recommend to expand the aims. Even, I suggest to describe them into a separate subsection "1.1.".

Response: Thank you very much. As recommended, we separated the section on aims and hypotheses, which are now presented under “1.1. Aims and hypotheses”, line 107. We did not expand the aims yet, because we did not examine any additional research questions and we wanted to write as concisely as possible. If we receive more specific information from Reviewer #2 on how and the extent to which we should expand the research aims, we will be happy to do so.

  1. Material and Methods: Section 2.1. is based on the description of patients with normal weight, overweight and obsesity. Perhaps, this section should be better renamed as "Participants and study design. " More details about how were participants recruited would be useful.

Response: Thank you for this suggestion. We now renamed the section to “Participants and study design” and added more details regarding recruitment. It now states: “Children at the age of 8 to 13 years were recruited from the population through advertisements in digital media, supermarkets, and citizen’s offices. In addition, study flyers and posters were placed in clinical institutions (e.g., University Medical Center, local pediatricians, outpatient weight-reduction programs). Interested children and their parents contacted the study team via telephone or email, and an appointment for a telephone screening was scheduled to evaluate inclusion and exclusion criteria.”, line 131 – 136.

  1. Data Analytic Plan: Data Analytic Plan is the main Data Analysis. Please, rename this section.

Response: As suggested, we renamed the section to “Data analysis”.

  1. Results: 3.1. Main characteristics of the total sample should be described.

Response: Following Reviewer #2’s comment, we now included information on sample characteristics for the total sample. It now states: “The total sample was 10.83 (± 1.79 SD) years old, included slightly more boys (n = 14, 56%) than girls (n = 11, 44%) and families with high socio-economic status (n = 15, 60%), as measured by the Winkler index [37]. All children had German nationality. Children’s mean BMIP was 70.42 (± 25.92 SD) […]”, lines 151 – 154.

  1. Discussion: At the beginning of the discussion section, the authors report that to the best of our knowledge, this is the first study investigating neural correlated processes in children with overweight and obesity. I recommend to expand it and describe into a Strengths section.

Response: As recommended, we expanded the first sentence of the Discussion section, now stating: “Unlike available research in adults with overweight and obesity [34], to the best of our knowledge, this study is the first to systematically investigate food-specific neural processes in children with overweight or obesity using quantitative EEG.”, lines 311 – 313. However, in order to reduce redundancy, we did not repeat this information in the strengths and limitation section of the Discussion. We hope that this is acceptable to the reviewer, and are happy to make the changes if further requested.